# Targeting Mitochondrial Dysfunction and Oxidative Stress to Prevent the Neurodegeneration of Retinal Ganglion Cells

**DOI:** 10.3390/antiox12112011

**Published:** 2023-11-17

**Authors:** Elisabetta Catalani, Kashi Brunetti, Simona Del Quondam, Davide Cervia

**Affiliations:** Department for Innovation in Biological, Agro-Food and Forest Systems (DIBAF), Università degli Studi della Tuscia, Largo dell’Università snc, 01100 Viterbo, Italy; kashi.brunetti@unitus.it (K.B.); simona.delquondam@unitus.it (S.D.Q.)

**Keywords:** redox homeostasis, mitochondria, neurodegeneration, retina, ganglion cells

## Abstract

The imbalance of redox homeostasis contributes to neurodegeneration, including that related to the visual system. Mitochondria, essential in providing energy and responsible for several cell functions, are a significant source of reactive oxygen and/or nitrogen species, and they are, in turn, sensitive to free radical imbalance. Dysfunctional mitochondria are implicated in the development and progression of retinal pathologies and are directly involved in retinal neuronal degeneration. Retinal ganglion cells (RGCs) are higher energy consumers susceptible to mitochondrial dysfunctions that ultimately cause RGC loss. Proper redox balance and mitochondrial homeostasis are essential for maintaining healthy retinal conditions and inducing neuroprotection. In this respect, the antioxidant treatment approach is effective against neuronal oxidative damage and represents a challenge for retinal diseases. Here, we highlighted the latest findings about mitochondrial dysfunction in retinal pathologies linked to RGC degeneration and discussed redox-related strategies with potential neuroprotective properties.

## 1. Introduction

The imbalance of reactive oxygen species (ROS) and reactive nitrogen species (RNS) turnover contributes to the progression of neurodegeneration. In the eye, low/mild ROS levels in cells/tissues could contribute to maintaining cellular homeostasis, signaling, and survival. In contrast, excessive ROS induces significant neuronal oxidative damage [1]. Therefore, balancing redox homeostasis contributes strongly to maintaining a proper level of ROS and RNS and, thus, healthy conditions. Mitochondria, essential in providing energy and other key cell functions, are a significant source of reactive species that can lead to the progression of pathologies, including those related to the visual system [2,3]. Still, ROS are crucial for communication between the mitochondria and the nucleus, although their overproduction could impair mitochondria, damaging mitochondrial DNA (mtDNA), proteins, lipids, and membranes. The resulting dysfunctional mitochondria could induce cellular apoptosis and neurodegeneration. Therefore, preserving proper mitochondrial homeostasis is essential for cells, including retinal neurons.

The retina is a part of the central nervous system, with a high metabolism and energy demands mainly satisfied by oxidative and glycolytic metabolism, resulting in ROS/RNS production and, possibly, accumulation. Within the retinal neurons, photoreceptors primarily require a high amount of energy. Indeed, photoreceptors are not only the first step in phototransduction but also represent a source of metabolic intermediates for the surrounding cells, being in close contact with the retinal pigment epithelium (RPE), which in turn is linked to the choroidal vasculature [4]. RPE supplies photoreceptors with plenty of glucose, which is crucial for the retinal neuron’s homeostasis and energy needs. Also, inside photoreceptors, mitochondria produce ATP using the fatty acids and ketone bodies that RPE supplies. In physiological conditions, different cells enact strategies to hamper metabolic impairments inside the retina [1]. Müller cells, one of the three types of retinal glial cells, fulfill the delicate role of supervising neurons’ necessities, including metabolic conditions. Activated Müller glia contribute to counterbalancing metabolic impairments. Still, its prolonged activation can lead to harmful conditions, such as inflammation and vascular defects. Also, retinal ganglion cells (RGCs), the inner-most retinal neurons, maintain homeostasis in the retina and regulate the retinal blood flow by secreting angiogenic factors, thus supervising the supply of nutrients to the entire retina.

Due to their central role in metabolic-related retinal pathologies, including age-related macular degeneration (AMD) and diabetic retinopathy (DR) [1,2,5,6,7], mitochondria appear frequently dysfunctional, contributing to the disease’s onset and progression. Also, age promotes mitochondrial impairment, and in this condition, mtDNA mutations accumulate more consistently than in the nuclear genome [2]. Of interest, mitochondrial dysfunction contributes to oxidative stress and impacts neurodegeneration in glaucoma [8]. All this evidence supports mitochondrial malfunctioning as a primary aspect to consider during eye/retinal pathologies. Noteworthy, during metabolic stress such as hyperglycemia, normal retinal physiology is dysregulated, with oxidative stress becoming a key aspect of the pathology [9,10]. Indeed, during diabetes, ROS accumulate, originating from different cellular events and cellular compartments, including mitochondria. DNA, protein modification, and lipid peroxidation are some consequences of altered redox signaling. In the pathogenesis of diabetic retinopathy, ROS production is an early event [11], and its accumulation impairs mitochondria, which in turn increases their own ROS production. This ROS accumulation triggers oxidative stress that induces inflammatory cytokine release, and it is well known that strict crosstalk between oxidative stress and inflammation in the etiology of DR exists. These events expose the retinal capillaries and neurons to cytosolic and mitochondrial ROS insults, leading to cellular apoptosis and degenerative processes. In these conditions, cellular defense control fails, resulting in severe deterioration. Mitochondria are primarily involved in these processes, contributing to ROS overproduction and other critical events, leading to the dysregulation of crucial pathways. For instance, mtDNA appears hypermethylated in hyperglycemic conditions, and its transcription is defective, leading to altered levels of the master transcription factor nuclear factor-erythroid 2 related factor 2 (Nrf2). It regulates the antioxidant response elements (AREs) system that mediates the transcription of antioxidant enzymes. Nrf2 also regulates the inflammatory mediator-NF-kB pathway and the expression of antioxidant genes that ultimately exert anti-inflammatory functions. Furthermore, some crucial antioxidant molecules appeared negatively regulated in mitochondria during DR, including the superoxide scavenging enzyme manganese superoxide dismutase (MnSOD, encoded with the *SOD2* gene) and antioxidant glutathione (GSH). Although several mechanisms involving mitochondrial dysfunction are established, how to counteract them is still mostly unclear.

In recent years, different authors have highlighted the importance of oxidative stress in the neurodegeneration of RGC, including those occurring in optic neuropathies and inherited retinal dystrophies, also suggesting future perspectives [12,13]. Of interest, autophagy induction through pharmacological intervention or genetic activation might be a valuable strategy for counteracting redox homeostasis dysregulation in major neurodegenerative diseases, including Alzheimer’s, Parkinson’s, and amyotrophic lateral sclerosis [14]. Given the centrality of this topic, the necessity of investigations on anti-oxidative biomarkers related to neurodegeneration, including retinal neurodegenerations, and the relevance of overviews that compare the results of different investigations to identify the key issues were emphasized [13]. Among them, mitochondrial dysfunction clearly emerged as a fundamental aspect [12,13].

In order to offer an additional and focused point of view, we review here the recent findings about mitochondrial dysfunction in retinal pathologies, mostly linked to RGC degeneration. In particular, we provide details on the mechanisms causing RGC damage and discuss the most valuable redox-related strategies suggested for their neuroprotection.

## 2. Mitochondrial Impairment Is Involved in RGC Degeneration

Due to their central role in sending action potentials to the brain, RGCs consume high energy for their metabolism and are particularly susceptible to mitochondrial dysfunction. RGCs mitochondrial impairment hampered ATP supply and was related to severe pathologies hitting the optic nerve and eventually causing glaucoma [15], or the entire neural retina, such as in DR. Not less noteworthy, mitochondrial dysfunction causes oxidative stress in RGCs, which contributes to neurodegeneration [16,17]. Under acute stress, RGCs can efficiently end damaged mitochondria, activating biogenesis to maintain energy homeostasis and the correct mitochondria number [15]. The balance between degradation and biogenesis of mitochondria is essential for healthy cells. On the contrary, when this balance falls, cells undergo snags. This is the case of a model of human stem cell differentiated RGCs (hRGCs) with the optineurin (a crucial actor for mitophagy) dominant mutation (E50K) [15]; this mutation was associated with normal-tension glaucoma [17]. In E50K hRGCs, restoring correct biogenesis through pharmacological treatment helps to balance energy homeostasis and counteract neurodegeneration [15]. In particular, in E50K hRGCs, inhibiting the tank-binding kinase 1 (TBK1) with the BX795 drug reduces cellular apoptosis, promotes mitochondrial biogenesis, increases mitochondrial mass, and minimizes mitochondrial swelling, which occurs when mitochondria increase their matrix volume to possibly produce more ATP. Also, in glaucomatous E50K hRGCs, activating the energy sensor AMPKα triggers, in turn, the mitochondrial biogenesis regulator PGC1α to maintain homeostasis. BX795 exerts its role independently from the AMPKα-PGC1α pathway. Interestingly, in non-mutant hRGCs in normal or under-stress conditions, BX795 treatment increased spare respiratory capacity and reduced apoptosis, suggesting an efficient neuroprotective strategy in non-genetic pathological conditions. Enhanced mitochondrial biogenesis was also observed in RGCs differentiated patient-specific human-induced pluripotent stem cells (hiPSCs) from MT-ND4-mutated Leber’s hereditary optic neuropathy (LHON)-affected patients [18]. The LHNO mutation reduces spare respiratory capacity, reducing mitochondria’s ability to supply energy and cell survival, promoting neurodegeneration. Furthermore, in this model, the expression of the antioxidant enzyme catalase appeared reduced, indicating higher oxidative stress and an imbalanced redox status. Additional studies have shown that MT-ND4 mutations induce elevated levels of oxidative stress that lead to dysfunctional and apoptotic RGCs [19]. Additionally, ROS inhibits the expression of the kinesin family member 5A (KIF5A) protein, causing an increment in the retrograde movement of the mitochondria in the axons. Also, the mix of the downregulation of KIF5A expression and the MT-ND4 mutation results in increased levels of apoptosis. The inhibition of the ERK1/2-Dynamin-related protein 1 (Drp1)–ROS axis was recently suggested as a potential therapeutic strategy to rescue RGC loss and counteract pathologically high intraocular pressure, a primary risk factor for glaucoma [20]. The detrimental effect of upregulation of p-ERK1/2 probably acts on Müller cells that, in turn, regulate the expression of p-Drp1 (Ser616) in RGCs. Drp1 is a mitochondrial protein that takes part in fission [21]. It was suggested that regulating its expression could help modulate mitochondrial ROS production and reduce RGC loss. Also, the Drp-1 protein was modulated via the A-Kinase anchoring protein 1 (AKAP1), a multifunctional mitochondrial scaffold protein that increases cell survival and regulates mitochondrial dynamics, bioenergetics, and mitophagy [22]. AKAP1 promotes mitochondrial elongation by regulating PKA/Drp1 anchoring, thus favoring mitochondrial good functioning. Furthermore, AKAP1 mediates mitochondrial bioenergetics by augmenting ATP synthesis and the mitochondrial membrane potential. Since oxidative stress and elevated intraocular pressure induce AKAP1 deficiency in RGCs and an increment of AKAP1 expression promotes RGC survival against oxidative stress, it was suggested that AKAP1 plays an essential role in mitochondrial preservation in RGCs during neurodegeneration induced by glaucoma. In RGCs, the loss of AKAP1 leads to an increment in Drp1 (Ser616) dephosphorylation, followed by mitochondrial fragmentation and loss [22,23]. On the contrary, phosphorylation of Drp1 mediated by AKAP1 promotes mitochondrial fusion and rescues RGCs from glaucoma insult [22]. In the glaucomatous DBA/2J retina and AKAP1^−/−^ mice, a significant increase in calcineurin (CaN) protein expression was detected, together with the dephosphorylation of Drp1 Ser637 [23]. In AKAP1^−/−^ mice, strong LC3 immunoreactivity was detected in RGC somas and axons, supported by the increment of LC3-II and the decrement of p62 levels, suggesting an enhancement in autophagosome production and, thus, the induction of autophagy/mitophagy. Furthermore, oxidative phosphorylation (OXPHOS) complexes (Cxs) deregulation was observed, with increased SOD2 protein expression, causing metabolic dysfunction and oxidative stress in the retina. Akt inactivation and Bim/Bax activation were also detected in AKAP1^−/−^ mice, contributing to glaucomatous neurodegeneration.

As well as in glaucoma neurodegeneration, RGCs are prone to deterioration in familial dysautonomia (FD), a disorder characterized by developmental and progressive neuropathies that affect the entire nervous system, also causing blindness [24]. FD is caused by a mutation in the *IKBKAP/ELP1* gene, which encodes the inhibitor of κB kinase complex-associated protein IKAP, also named ELP1, involved in elongation and demanded the translation of codon-biased genes. In a mouse model of FD blindness, it was observed that the loss of IKAP caused progressive degeneration of RGCs with a progressive loss of mitochondrial membrane integrity, membrane potential, function, and thus ROS dysregulation. Interestingly, the other retinal neurons, including Müller glial, bipolar, amacrine, and photoreceptor cells, remained mainly uninjured, though with damaged mitochondria. This evidence supports that RGCs are highly vulnerable to mitochondrial dysfunction, probably due to their high energy demand and unique morphology. Recently, a case report described the thinning of the retinal nerve fiber and ganglion cell layers and decreased mitochondrial function in a 17-year-old patient [25]. This report associates the *SIRT3* (*sirtuin 3*) gene mutation with mitochondrial optic neuropathy. SIRT3 is a well-known regulator of mitochondrial metabolism [26]. Supporting the concept that mitochondria biogenesis is crucial in maintaining homeostasis, it was observed that intravitreal transplanted iPSC-MSCs might donate functional mitochondria to RGCs in a mouse model of Leigh’s disease, contributing to protecting them from cell degeneration and death [27]. Furthermore, iPSC-MSCs reduce abnormal activation of Müller cells and inflammation by reducing neuroinflammatory cytokines such as TNF α, MIP−1g, GM-CSF, IL-5, IL-17, and IL-1 β, protecting RGCs from degeneration and loss. This evidence underlines a link between mitochondrial dysfunction, inflammation, and neurodegeneration. Glia neuroinflammation is a significant contributor to glaucoma. During elevated intraocular pressure conditions, an upregulation of TLR4 and IL-1β expression in Müller glia end feet was evident, both in the human glaucomatous retina and in the DBA/2J mouse that mimics human glaucoma [28]. At the same time, a significant decrement of apolipoprotein A-I binding protein (AIBP; gene name *Apoa1bp*) was detected in RGCs, leading to spatial vision dysfunction but not severe optic nerve damage. In Apoa1bp^−/−^ mice, AIBP deficiency activates mitochondrial fragmentation, mitochondrial cristae depletion, and energy production dysregulation, resulting in dysfunctional Müller glia and inflammatory conditions. AIBP deficiency impairs mitochondrial dynamics and decreases Cxs protein expression in the retina. In particular, in Apoa1bp^−/−^ RGC somas, mitochondrial fragmentation and reduced ATP production in RGCs were detected. Furthermore, an AIBP role was also suggested in the inner retina, affecting RGC dendrites in the IPL during glaucomatous neurodegeneration. In RGCs and the inner retina, AIBP deficiency contributes to oxidative stress, reducing SIRT3 and SOD2 amounts and increasing phospho-p38, a stress-signaling player, and ERK1/2. On the contrary, the administration of AIBP promotes RGC and inner retinal neuron survival and inhibits oxidative stress signaling and inflammatory responses in mice, which could result in neuroprotection. This evidence suggests that AIBP may have therapeutic prospects for treating glaucoma, blocking neuroinflammation, and acting on mitochondrial functions.

## 3. Mitochondria Homeostasis as a Target against RGC Degeneration

Undeniably, mitochondrial surveillance is a central point in neuroprotection. Mitochondria surveillance includes fusion, fission, degradation of damaged mitochondria (mitophagy), and the biogenesis of new mitochondria [15]. Healthy cells renovate their set of mitochondria through biogenesis, which is also a process that could substitute damaged or dysfunctional mitochondria [29]. Still, when misregulated, mitochondria biogenesis leads to detrimental effects observed in aging, metabolic disease, cancer, and neurodegeneration, including Alzheimer’s disease, Parkinson’s disease, Huntington’s disease, and amyotrophic lateral sclerosis. Due to its crucial role, mitochondria biogenesis is a therapeutic target in neurodegenerative diseases, including those involving the visual system. A particular link between mitochondrial biogenesis and Nrf1 was highlighted during retinal development [30]. Indeed, RGCs are very sensitive to Nrf1 deletion, more so the newly differentiated one than the terminally differentiated one, suggesting a role in neurite outgrowth and metabolic reprogramming. This evidence supports Nrf1 as a crucial element involved in dysfunctional mitochondrial biogenesis that could contribute to the pathology and disease progression in neurodegenerative diseases, and thus, it could be a target for therapeutics.

In support of mitochondria as a therapeutic target, liver-isolated mitochondria were transplanted into RGCs after optic nerve crush, and axonal outgrowth was investigated [31]. This mitotherapy approach revealed that transplanted healthy mitochondria do not hamper retinal redox homeostasis. On the contrary, transplanted mitochondria preserved retinal electrophysiological activity, enhancing both a- and b-wave amplitudes and protecting RGCs from cell death. Since mitochondrial respiratory dysfunction is the primary cause of glaucoma, these findings offer a potential in vivo regenerative approach for neurodegenerative retinal pathologies. Rescuing mitochondrial functionality and decreasing ROS could preserve RGCs from death and can be achieved with gene therapies [32]. The nuclear yeast gene *NADH-quinone oxidoreductase* (*NDI1*) was optimized (ophNdi1) and suggested as a new gene therapy for ocular disorders with mitochondrial deficits. Adeno-Associated Virus (AAV)-ophNdi1 was tested in a rotenone-induced murine model of optic neuropathy. Results revealed a reduction in RGC loss and an increment in visual function. Furthermore, AAV-ophNdi1 was tested in complex I deficient patient-derived fibroblasts, which expressed the G11778A mutation in ND4 that causes complex I dysfunction and LHON, revealing beneficial effects, an increment in oxygen consumption, and ATP production rates.

Acting on cellular homeostasis when cells are still alive and, thus, preventing degeneration is a crucial point. Recently, it was observed that the early response of RGCs to diabetes implicates specific morpho-functional deficits long before cell death [33]. Most importantly, it underlined the relevance of neuronal protection in the early phases of diabetic retinopathy, when RGC morphology can still be preserved and homeostasis maintained [34]. Using Thy1-green fluorescent protein (GFP)-M transgenic mice, which express GFP in several RGCs, the early response of RGCs to diabetes was described, also after the neuroprotective somatostatin (SRIF) analog octreotide (OCT) treatment. OCT preferentially binds somatostatin subtype receptors (sst) 2 and 5 and has moderate affinity for sst3 [33,35,36]. In the mouse retina, sst5 receptors are expressed via RGCs, while sst2 has been detected in rod bipolar cells and selected populations of amacrine cells [37]. Therefore, OCT may protect RGCs from hyperglycemia-induced damage acting directly on sst5. Accordingly, during glaucoma, the selective activation of sst5 is protective for RGCs [38]. Furthermore, OCT could act indirectly on RGCs by coupling to sst2, expressed via bipolar cells and amacrine cells. Currently, SRIF analogs represent a treatment option in the field of ophthalmology, including DR [39]. Among SRIF receptors in RGCs, also sst1 and sst4 were observed in rats (in dendrites and cell bodies) and mice RGCs [37]. Functionally, activating sst4 by SRIF or its agonist L-803087 provides potential targets to reduce intracellular Ca^2+^ levels in RGCs. It was also found that T-type Ca^2+^ currents were reduced by the administration of the sst5 receptor-specific agonist L-817818 [40]. Most importantly, the L-817818 agonist increases RGC survival, reducing apoptosis in a rat glaucoma model [38]. Of interest, L-817818 treatment inhibited reactive oxygen species and malondialdehyde formation, thus reducing retinal oxidative stress and, likely, cell injury. Sst5 activation also reduced mitochondrial dysfunction, which resulted in RGC’s protection. Still, even if SRIF analog treatment effectively protects RGCs during metabolic stress, it presents some challenges. Indeed, SRIF analog OCT treatment of hyperglycemic retinas fully rescues RGC morphometric parameters, depending on the complexity of the dendritic tree, or, on the contrary, increases the effects of hyperglycemia [33]. Remarkably, RGCs emerged as the first retinal neurons to display functional deficits after diabetes onset. These deficits can be appreciated as early as 11 days after the first recording of a hyperglycemic status, as assessed with blood glucose levels. In particular, electrophysiological analysis of the photopic negative response (PhNR) amplitude and pattern electroretinography (PERG) related to RGC activity indicate that the intravitreal treatment with the somatostatin analog OCT counteracts amplitude decreases. These observations indicate that early neuroprotective intervention may efficiently balance the first functional alterations due to diabetes. Furthermore, OCT recovers some RGC subtypes and induces adaptive changes in others. Indeed, based on electrophysiological and morphological evidence, OCT provides complete protection from hyperglycemia-induced damage in some RGCs. However, in other RGCs, OCT may induce changes that promote adaptation to hyperglycemic metabolic stress, contributing to their regular functional activity. These changes will likely include a reduction in energy expenditure through autophagic process enhancement as a protection strategy, as highlighted in ex vivo retinal explants treated with OCT [41]. In addition, recently, it was demonstrated that stimulating the oxidative stress response and enhancing mitochondrial activity during hyperglycemia result in neuroprotection of the visual system [42,43]. In particular, using a *Drosophila melanogaster* high glucose-induced eye damage system that resembles the hyperglycemic phenotype of higher vertebrates and models the early phases of DR [44], we found ROS accumulation and upregulation of the main oxidative stress response factors, including superoxide dismutases, catalases, glutathione-S-transferase (GstD1), and cap ’n’ collar isoform C (CncC), the homolog of mammalian Nrf2 [42,43]. In normal conditions, Nrf2 in the retina is expressed via various retinal cell types, including Müller and ganglion cells, and its dysregulation is associated with a severely altered antioxidant response mechanism and the reduction in RGC integrity and survival [45,46]. To confirm Nrf2 dysregulation in hyperglycemic flies, a decrement of kelch-like ECH-associated protein 1 (Keap1), which regulates Nrf2 transcriptional activity, was detected [42]. Most importantly, morphologically damaged mitochondria were identified in fly eye tissue sections, and impaired mitochondrial activity was observed in the homogenate of the fly head extracts. In addition, an upregulation of the main transcriptional effectors of unfolded protein response (UPR) in nervous system cells was demonstrated. Activating transcription factor 4 (ATF4) is an oxidative stress-inducible signal activated in neurons via the perturbation of mitochondria that mainly controls autophagy, protein folding, redox balance, and apoptosis [47,48]. Accordingly, ATF4 is overexpressed in degenerating retina models, including diabetic retinopathy, and its downregulation reduces the mitochondrial–endoplasmic reticulum (ER) stress response, preventing cell death and the impairment of vision function [48]. Interestingly, administering to the flies the natural antioxidant compound plumbagin as a drug, Nrf2 (CncC) and GstD1 were additionally upregulated but not keap1 expression. The modulation of Nrf2 (CncC)/Keap1/GstD1 suggests that this signaling might target the neuronal response to ROS damage during hyperglycemia [42]. We also observed in flies that the antioxidant plumbagin contributes significantly to increasing mitochondria morphology and activity and reduces ATF4 levels, exerting a neuroprotective role by inhibiting mitochondria-related UPR signals under prolonged ER stress conditions. Also, this neuroprotective action of plumbagin involves the downregulation of HSP70 expression, suggesting a modulation in the stress-induced recruitment of molecular chaperones to recover fly retinal homeostasis after oxidative damage. Regarding signaling, plumbagin may act additively and synergistically, inhibiting the mitochondrial-ER stress response and UPR signals, which prevents ROS-induced neuronal impairment and eye injury. All this evidence confirmed the positive effects of antioxidant substances against high glucose-induced eye damage, suggesting a contribution to mitochondrial functionality. As shown in Figure 1, the mitochondrial activity in adult flies was negatively affected by high glucose feeding.

In particular, the MTT reductive ability significantly decreased in homogenate head extracts of flies raised on food added with a 30% high sucrose diet (HSD 30%) when compared to a standard diet (STD) containing 5–9% of sucrose. The major decrement was obtained with a very high sucrose diet (HSD 40%). Since damage to the visual system increases with the increment of body sugar [44], these findings are in line with the notion that hyperglycemia primarily affects mitochondrial homeostasis in retinal neurons. A reduction in ROS and the upregulation of the retina’s glutathione system resulted after treating hyperglycemic flies with the natural product Lisosan G, a powder of bran and germ of grain (Triticum aestivum) [43]. Also, Lisosan G reduced morphological damage and apoptosis, restoring autophagic flux balance, thus supporting an antioxidant effect and neuroprotection in the retina and possibly preserving mitochondrial activity. The efficacy of antioxidants in improving mitochondrial function and neuroprotection in the retina was also demonstrated using resveratrol [49]. Resveratrol increased mitochondrial function, upregulating Ampk/Sirt1/Pgc1α and downregulating the Akt/mTOR pathway in an age-related retinal neuropathy zebrafish retina. Recently, a set of in vivo and in vitro experiments confirmed the efficacy of resveratrol against AMD. In particular, Nguyen and colleagues [50] have developed a nanomedicine strategy using R@PCL-T/M NP nanotherapeutics to improve the simultaneous delivery to the retina of resveratrol and metformin drugs. Through some elegant experiments, they demonstrated that R@PCL-T/M NP nanotherapeutics are highly biocompatible with retinal cells without compromising the robust antioxidant effect of resveratrol. Indeed, they observed that R@PCL-T/M NP treatment attenuates ROS production, exerts anti-inflammatory effects in vitro, and displays antiangiogenic properties in vitro and in vivo. This innovative approach enhanced the retinal permeability of the drugs in vivo, which helps with the simultaneous delivery of resveratrol and metformin, even to inner segments of the retina as the RPE region. Altogether, these findings are promising for developing pharmacological nanoformulations targeting retinal cells during pathological conditions, including AMD.

Collectively, all these findings elucidate the close relationship between mitochondrial dysfunction and RGC degeneration and suggest possible targets to counteract visual system pathologies (Figure 2). These results also emphasize that targeting mitochondrial functionality is a promising strategy for preventing oxidative stress damage in neurodegenerative retinas resulting from metabolic impairment and confirm the need to develop innovative strategies for effective therapeutic interventions.

## 4. Concluding Remarks

Correct mitochondrial functionality is crucial to preventing retinal neurodegeneration. Several mitochondrial retinopathies exist [51,52], and some of them are associated with non-retinal rare pathologies such as mitochondrial encephalomyopathy with lactic acidosis and stroke-like episodes (MELAS) and maternally inherited diabetes and deafness (MIDD), and that could even help their diagnosis [52]. These retinopathies could result from specific mitochondrial variants [51,52], including the m.3243G>A variant, and mutations in the fusion/mitochondrial shaping protein OPA1, encoded with a nuclear gene on chromosome 3q29. Also, several neurodegenerative diseases that display oxidative stress and changes in ocular physiology, such as Alzheimer’s disease, Parkinson’s disease, Huntington’s disease, multiple sclerosis, and muscle dystrophy, are associated with mitochondrial dysfunctions in neurons [53,54,55,56,57,58,59]. Thus, the dual targeting of mitochondria and ROS could be an excellent strategy to prevent, counteract, and/or delay retinal neurodegenerative pathologies, including those related to the degeneration of RGCs, such as glaucoma and DR (Figure 3).

Indeed, it was demonstrated that preventing mitochondrial fragmentation represents a chance to protect retinas from various malfunctions affecting cellular metabolism, cellular respiration, and apoptosis that participate in the development and progression of DR. In this context, the antioxidant treatment represents a chance for neuroprotection, improving mitochondrial morphology and functions, and counteracting retinal pathologies. Nevertheless, there are challenges in using antioxidants, such as over-limiting ROS, as they are essential for normal cellular processes, and the low bioavailability of conventional antioxidants, especially in the retina, due to their isolation from the main bloodstream. Therefore, there is a need for new advances in this direction, also by using modern technologies such as, for instance, nanoformulations for drug delivery [50]. In considering targeting mitochondrial dynamics to implement sound therapeutic strategies, it is crucial to intervene promptly before ROS irreversibly harms cells [12]. Although this is a critical aspect and not easy to accomplish, targeting mitochondrial dynamics appears to be a challenge for future approaches. The autophagy machinery manipulation offers an interesting perspective against neurodegeneration. Indeed, it was suggested that over-activation of autophagy could be a potential strategy for protection against oxidative stress and the prevention of neurodegenerative disorders [14,60,61].

This evidence underlines that each progress in molecular and functional investigations represents a great opportunity for intervention against visual damage related to oxidative stress.

## Figures and Tables

**Figure 1 antioxidants-12-02011-f001:**
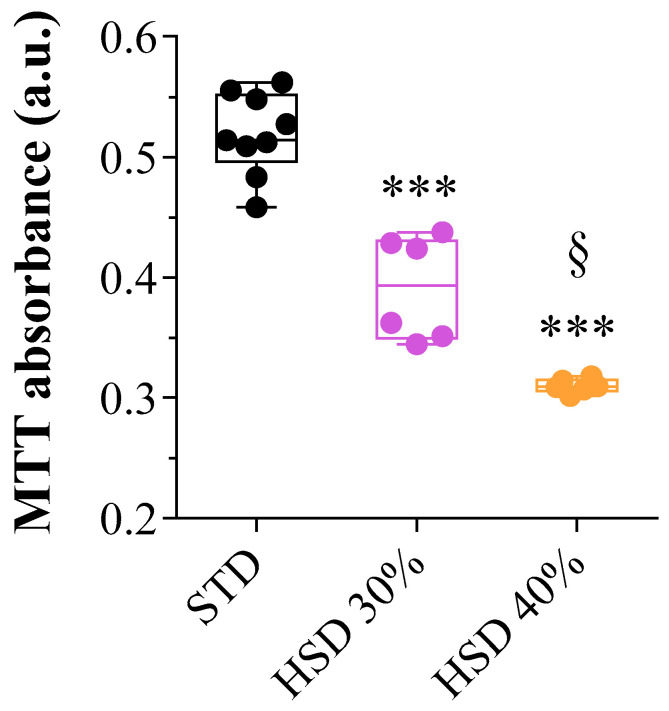
Measurements of mitochondrial activity with MTT (3-(4,5-dimethylthiazol-2-yl)-2,5-diphenyltetrazolium bromide) assay absorbance in adult *D. melanogaster* head extracts after ten days of feeding with a standard diet (STD, 5–9% sucrose), a high sucrose diet (HSD) of 30%, and a very HSD of 40%, according to the method previously published [43]. Data representing ca. 100 heads were obtained from at least 5 independent experiments. The statistical significance was evaluated using a one-way ANOVA followed by the Tukey post-test. The results were expressed as means ± SEM. a.u.: arbitrary units. *** *p* < 0.0001 vs. STD, and § *p* < 0.01 vs. HSD 30%.

**Figure 2 antioxidants-12-02011-f002:**
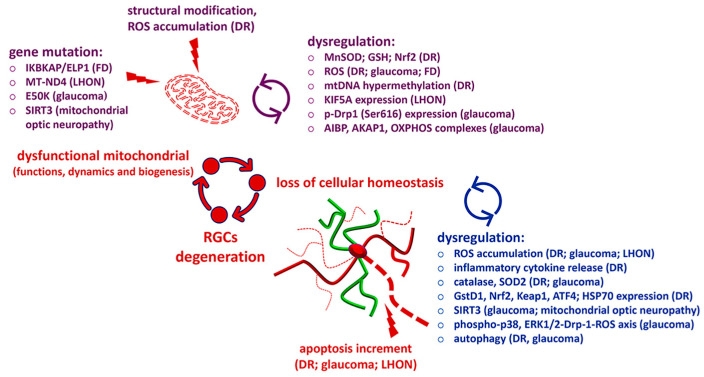
Latest findings about mitochondrial dysfunction causing retinal ganglion cell (RGC) degeneration and the main factors involved. Reactive oxygen species (ROS); diabetic retinopathy (DR); familial dysautonomia (FD); Leber’s hereditary optic neuropathy (LHON); inhibitor of kappa light polypeptide gene enhancer in b cells; kinase complex-associated protein/elongator acetyltransferase complex; subunit 1 (IKBKAP/ELP1); mitochondrially encoded NADH:ubiquinone oxidoreductase core subunit 4 (MT-ND4); optineurin dominant mutation (E50K); sirtuin 3 (SIRT3); manganese superoxide dismutase (MnSOD); glutathione (GSH); nuclear factor-erythroid-2-related factor 2 (Nrf2); mitochondrial DNA (mtDNA); kinesin family member 5A (KIF5A); phospho-dynamin-related protein 1 (p-Drp1); apolipoprotein A-I-binding protein (AIBP); A-Kinase-anchoring protein 1 (AKAP1); oxidative phosphorylation (OXPHOS); superoxide dismutase 2 (SOD2); glutathione-S-transferase (GstD1); kelch-like ECH-associated protein 1 (Keap1); activating transcription factor 4 (ATF4); heat shock protein 70 (HSP70); and extracellular signal-regulated kinase 1/2 (ERK 1/2).

**Figure 3 antioxidants-12-02011-f003:**
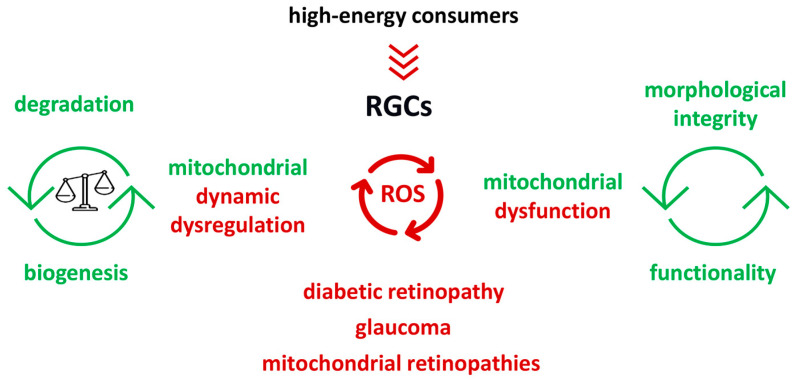
Oxidative stress is a hallmark of neurodegenerative retinal diseases. Reactive oxygen species (ROS) accumulation affects mitochondria that undergo dynamic dysregulation and dysfunction. Retinal ganglion cells (RGCs) are high-energy consumers and are particularly sensitive to oxidative stress damage and mitochondrial impairment.

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
