# Peer review of "Targeting Mitochondrial Dysfunction and Oxidative Stress to Prevent the Neurodegeneration of Retinal Ganglion Cells"

_antioxidants, 2023, doi:10.3390/antiox12112011_

Round 1
Reviewer 1 Report
Comments and Suggestions for Authors
The current manuscript summarizes the current status about the targeting mitochondria dysfunction and oxidative stress to prevent the neurodegeneration of retinal ganglion cells. Although the topic is interesting in its scientific field, there are some issues that require the authors’ attention to improve the quality of this particular manuscript before further consideration for publication in a high-quality journal “Antioxidants”.
Specific comments:
1. The authors should carefully clarify the differences in the academic contribution points between the current manuscript and some earlier reports (please refer to the following papers: #1 DOI: 10.1007/978-3-030-27378-1_84 & #2 10.3390/antiox10050694 & #3 DOI: 10.3390/antiox10101538).
2. The authors should consider the inclusion of some figures or diagrams to clearly illustrate the key concepts and mechanisms.
3. In this review, only Figure 1 presents the detrimental effects of high glucose feeding on mitochondrial activity in Drosophila. Given that the main focus of this contribution is to summarize the findings related to mitochondrial dysfunction in retinal pathologies, the authors should pay particular attention to the area of RGC degeneration.
4. Since the balance of the redox homeostasis in retinal cells is the focus of this review, the authors should illustrate some specific strategies that can be used to balance the redox homeostasis of retinal cells.
5. As mentioned in Section 2 “Mitochondrial impairment is involved in RGC degeneration”, various pathways and mechanisms are described. However, the authors are highly recommended to consider the inclusion of diagrams illustrating the pathways and mechanisms in RGC degeneration due to mitochondrial impairment.
6. As stated by the authors, the efficacy of antioxidants in improving mitochondria function and neuroprotection in the retina was also demonstrated using resveratrol [55]. In fact, a recent study has also demonstrated the potential of resveratrol to boost the antioxidant defense system in the retina and to enhance therapeutic efficacy of macular degeneration (DOI: 10.1021/acsnano.2c05824). In order to balance scientific viewpoint and update article content, the authors are highly recommended to consider the inclusion of this supportive case study in the reference list.
Author Response
Reviewer comment 1
- The authors should carefully clarify the differences in the academic contribution points between the current manuscript and some earlier reports (please refer to the following papers: #1 DOI: 10.1007/978-3-030-27378-1_84 & #2 10.3390/antiox10050694 & #3 DOI: 10.3390/antiox10101538).
Authors replay
We thank the Reviewer for her/his attentive comments that gave us the chance to improve our manuscript. As the Reviewer suggested we have now better clarified our contribution to the field (Line 85-100, red text) “In recent years different authors have highlighted the importance of oxidative stress in the neurodegeneration of RGC, as those occurring in optic neuropathies and inherited retinal dystrophies, also suggesting future perspectives [12, 13]. Of interest, autophagy induction through pharmacological intervention or genetic activation might be a valuable strategy for counteracting redox homeostasis dysregulation in major neurodegenerative diseases, including Alzheimer's, Parkinson's, and amyotrophic lateral sclerosis [14]. Given the centrality of this topic, the necessity of investigations on anti-oxidative biomarkers related to neurodegeneration, including retinal neurodegenerations, and the relevance of overviews that compare the results of different investigations to identify the key issues were emphasized [13]. Among them, mitochondrial dysfunction clearly emerged as a fundamental aspect [12, 13].
In order to offer an additional and focused point of view, we review here the recent findings about mitochondrial dysfunction in retinal pathologies, mostly linked to RGCs degeneration. In particular, we provide details on the mechanisms causing RGCs damages and discuss the most valuable redox-related strategies suggested for their neuroprotection.”
Reviewer comments 2 and 3
- The authors should consider the inclusion of some figures or diagrams to clearly illustrate the key concepts and mechanisms.
- In this review, only Figure 1 presents the detrimental effects of high glucose feeding on mitochondrial activity in Drosophila. Given that the main focus of this contribution is to summarize the findings related to mitochondrial dysfunction in retinal pathologies, the authors should pay particular attention to the area of RGC degeneration.
Authors replay
We agree with this concern. Also, in agreement with the concern of Referee 2, in the revised manuscript we have now included a new figure at the end of section 3 (Figure 2), displaying the latest evidence that links mitochondrial dysfunctions with retinal pathologies. Furthermore, as suggested by the Reviewer, the illustration includes the main aspects that participate in the degenerative processes of retinal ganglion cells. We do hope that the Reviewer will appreciate this.
Reviewer comment 4
- Since the balance of the redox homeostasis in retinal cells is the focus of this review, the authors should illustrate some specific strategies that can be used to balance the redox homeostasis of retinal cells.
In section 3 “Mitochondria homeostasis as a target against RGCs degeneration” some specific strategies to counteract retinal pathologies linked to mitochondrial dysfunction have been reported. They are described to be active in rebalancing redox homeostasis, including the use of antioxidant molecules that in some cases improve mitochondrial activity. In addition, in order to better focus on this aspect, we have now improved the discussion by adding new results (lines 343-361, red text, see authors replay point 6) that suggest innovative strategies for drug delivery in vivo. Finally, the new Figure 2 provides an overview of several factors involved in redox homeostasis in retinal cells.
Reviewer comment 5
- As mentioned in Section 2 “Mitochondrial impairment is involved in RGC degeneration”, various pathways and mechanisms are described. However, the authors are highly recommended to consider the inclusion of diagrams illustrating the pathways and mechanisms in RGC degeneration due to mitochondrial impairment.
Thank you to the Reviewer for the appropriate observation. See the author's replay to the reviewer's comments 2-3
Reviewer comment 6
- As stated by the authors, the efficacy of antioxidants in improving mitochondria function and neuroprotection in the retina was also demonstrated using resveratrol [55]. In fact, a recent study has also demonstrated the potential of resveratrol to boost the antioxidant defense system in the retina and to enhance therapeutic efficacy of macular degeneration (DOI: 10.1021/acsnano.2c05824). In order to balance scientific viewpoint and update article content, the authors are highly recommended to consider the inclusion of this supportive case study in the reference list.
Thank you to the Reviewer for the correct suggestion, we agree with its point of view. We have taken into account its suggestion, and we have now added and commented on the suggested reference in the text, lines 343-361, red text. “Recently, a set of in vivo and in vitro experiments confirmed the efficacy of resveratrol against AMD. In particular, Nguyen and colleagues [51] have developed a nanomedicine strategy using R@PCL-T/M NP nanotherapeutics to improve the simultaneous delivery to the retina of resveratrol and metformin drugs. Through some elegant experiments, they demonstrated that R@PCL-T/M NP nanotherapeutics are highly biocompatible with the retinal cells without compromising the robust antioxidant effect of resveratrol. Indeed, they observed that R@PCL-T/M NP treatment attenuates ROS production, exerts anti-inflammatory effects in vitro, and displays antiangiogenic properties in vitro and in vivo. This innovative approach enhanced the retinal permeability of the drugs in vivo, which helps with the simultaneous delivery of resveratrol and metformin, even to inner segments of the retina as the RPE region. Altogether, these findings are promising for developing pharmacological nanoformulations targeting retinal cells during pathological conditions, including AMD.
Collectively all these findings elucidate the close relationship between mitochondrial dysfunction and RGCs degeneration and suggest possible targets to counteract visual system pathologies (Figure 2). These results also emphasize that targeting mitochondria functionality is a promising strategy for preventing oxidative stress damage in neurodegenerative retinas resulting from metabolic impairment and confirm the need to develop innovative strategies for effective therapeutic interventions.”
Reviewer 2 Report
Comments and Suggestions for Authors
I read the manuscript entitled “Targeting mitochondria dysfunction and oxidative stress to prevent the neurodegeneration of retinal ganglion cells” by Catalani and colleagues with great interest. The manuscript is scientifically sound and it delas with the most relevant diseases of the retina and optic nerve, diabetic retinopathy and glaucoma. I have only minor remarks.
In my opinion, a weakness of the manuscript is the low number of figures. I suggest to include some illustrations on mitochondrial dysfunction in the pathophysiology of diabetic retinopathy and glaucoma including the important molecules involved.
Comments on the Quality of English LanguageThe authors should check the grammar: E.g., within a sentence it is “optineurin” not “Optineurin”. Or “familial dysautonomia” not “Familial dysautonomia”.
Author Response
Reviewer comment 1
- In my opinion, a weakness of the manuscript is the low number of figures. I suggest to include some illustrations on mitochondrial dysfunction in the pathophysiology of diabetic retinopathy and glaucoma including the important molecules involved.”
Authors Reply
Thank you to the Reviewer for the positive opinion concerning the manuscript.
In agreement with the Reviewer's point of view, we have now inserted the new Figure 2 at the end of section 3, It displays the latest evidence that links mitochondrial dysfunctions with retinal pathologies, including diabetic retinopathy and glaucoma. Furthermore, as suggested by the reviewer, the illustration includes the main factors that participate to the degenerative processes of retinal ganglion cells.
Reviewer comment 2
- Comments on the Quality of English Language
The authors should check the grammar: E.g., within a sentence it is “optineurin” not “Optineurin”. Or “familial dysautonomia” not “Familial dysautonomia”.
Authors Reply
The grammar has been now further checked. We apologize for the mistakes that have been now corrected.
Round 2
Reviewer 1 Report
Comments and Suggestions for Authors
The revised version has adequately addressed most of the critiques raised by this reviewer and is now suitable for publication in "Antioxidants".